# CAREFL: Context-Aware Recognition of Emotions with Federated Learning

## Abstract

Emotion recognition from images is a challenging task due to its dependence on subtle visual cues and contextual information. Recent advances in Vision-Language Models (VLMs) have demonstrated strong performance in this domain. Still, they are often limited by their large computational footprint and the privacy concerns associated with centralized training. To address these challenges, we propose CAREFL (Context-Aware Recognition of Emotions with Federated Learning), a framework for efficient emotion recognition. CAREFL combines large VLMs, specifically LLaVA 1.5, for generating rich contextual descriptions with lightweight small VLMs, SMOLVLM2, fine-tuned under a federated learning setup using Quantized Low-Rank Adaptation. This design enables accurate, privacy-preserving, and resource-efficient training on edge devices. Although this work evaluates CAREFL in the context of emotion recognition, the framework is general by design: leveraging VLMs, it can also be fine-tuned for a wide range of multimodal description and classification tasks beyond emotion analysis. Extensive experiments demonstrate that CAREFL outperforms state-of-the-art baselines, achieving up to 96.49% mAP and 50.36% F1-score, surpassing heavier models such as GPT-4o, LLaVA, and EMOTIC. An ablation study further confirms the contribution of contextual enrichment, prompt design, and quantization in enhancing performance. The results show that federated fine-tuning of lightweight VLMs, when guided by contextual reasoning from large-scale models, provides a practical and scalable solution for emotion recognition in privacy-sensitive and resource-constrained environments.

## 1 Introduction

Understanding human emotions from visual data is central to applications in education, healthcare, and human-computer interaction (Pescarin & Pandiani, 2022; Saffar et al., 2023; Cambria, 2016). Recent advances in Visual-Language Models (VLMs) have demonstrated strong capabilities in recognizing subtle affective cues by leveraging both visual and contextual information (Lu et al., 2024; Liu et al., 2023). However, these gains come at a cost: Large VLMs such as GPT-4V or LLaVA demand significant computational memory and rely on centralized training with user data collected on a single server (Vajrobol et al., 2024). For emotion recognition and similar sensitive tasks, this raises serious concerns about privacy, scalability, and real-world deployment.

Federated learning (FL) provides a natural solution, enabling models to be collaboratively trained across distributed clients without sharing raw data (McMahan et al., 2023). Yet applying FL to multimodal VLMs is far from straightforward (Kairouz et al., 2021). Lightweight models, like SMOLVLM (Marafioti et al., 2025), often underperform in capturing the rich contextual signals needed for emotion understanding, while directly federating large VLMs is impractical due to prohibitive memory and communication costs. Moreover, most prior FL research for vision tasks has focused on efficiency or robustness, while ignoring the role of contextual reasoning that is critical for affective understanding (Barrett et al., 2011; Kosti et al., 2017).

In this work, we introduce CAREFL (Context-Aware Recognition of Emotions with Federated Learning), a framework that bridges this gap by combining the contextual power of large VLMs with the efficiency of lightweight small VLMs (SVLMs) in a federated setting. CAREFL operates in two phases: (i) an offline context generation phase, where a large VLM (LLaVA 1.5) produces de-

scriptive contextual captions for each image; and (ii) a federated fine-tuning phase, where an SVLM model (SMOLVLM2) is adapted with quantized low-rank adapters (QLoRA) on local devices, transmitting only compressed updates to the server. This design enables CAREFL to preserve privacy, efficiency, and contextual richness simultaneously.

Our contributions are threefold:

- We propose CAREFL, a framework that integrates context generation from large VLMs with lightweight SVLM fine-tuning under federated learning.

- We design an efficient QLoRA adaptation update scheme, which reduces memory footprint and communication overhead while maintaining high performance.

- We conduct extensive experiments on EMOTIC and CAER-S, showing CAREFL outperforms centralized and federated baselines in mAP and F1-score, while being significantly more efficient. Ablations further validate robustness across non-IID client splits, LoRA ranks, quantization levels, and aggregation methods.

By explicitly incorporating context into federated multimodal learning, CAREFL advances the design of scalable, privacy-preserving emotion recognition systems and opens a new path for deploying VLM-based reasoning under realistic distributed constraints.

## 2 RELATED WORK

Recent advances in artificial intelligence have improved emotion recognition, but accurately capturing complex emotional states in diverse, decentralized settings remains challenging. This section reviews progress in context-aware emotion recognition, the use of VLMs, and the role of FL.

### 2.1 FEDERATED LEARNING FOR EFFICIENT ADAPTATION

Federated learning (FL) enables collaborative model training without centralizing user data, making it especially relevant for privacy-sensitive domains (McMahan et al., 2023). Early FL approaches focused on vision models with FedAvg and its variants (Luo et al., 2025; Cui et al., 2024; Zeng et al., 2024). More recent work has explored parameter-efficient fine-tuning in FL. Works like FLoRA reduces communication costs by restricting updates to low-rank adapters (LoRA) (Nguyen et al., 2024). Similarly, FedPrompt (Zhao et al., 2023) leverages prompt tuning to adapt models in federated settings. While effective for efficiency, these works are limited to unimodal vision or language tasks, and don't address the role of contextual reasoning in multimodal learning.

### 2.2 VISION-LANGUAGE MODELS FOR CONTEXT-AWARE EMOTION RECOGNITION

Large VLMs such as CLIP, LLaVA, and GPT-4V have achieved strong performance in tasks that require joint visual-text reasoning (Hu et al., 2023; Etesam et al., 2024a; Lu et al., 2024; Yang et al., 2023). Recent studies have explored their use for emotion recognition and contextual inference, showing that context often improves accuracy on ambiguous affective cues. For example, Xenos et al. (2025) combines the use of LLaVA and CLIP, enriching emotion understanding by combining VLM-generated captions to improve the classification of emotions. However, these approaches operate in centralized settings, requiring large-scale resources and direct access to sensitive data (Etesam et al., 2024b; Baldassini et al., 2024).

While prior works mainly explore zero-shot FER, VLM + LLM frameworks, or emotion reasoning through prompting (Cheng et al., 2024; Xie et al., 2024), they often rely on large models directly at inference time, which limits efficiency on edge devices. Our two-phase pipeline instead uses LLaVA offline for context generation and delegates emotion inference to a fine-tuned lightweight SVLM. This separation allows us to harness the semantic power of large VLMs while ensuring computational and memory efficiency for real-world, resource-constrained deployment.

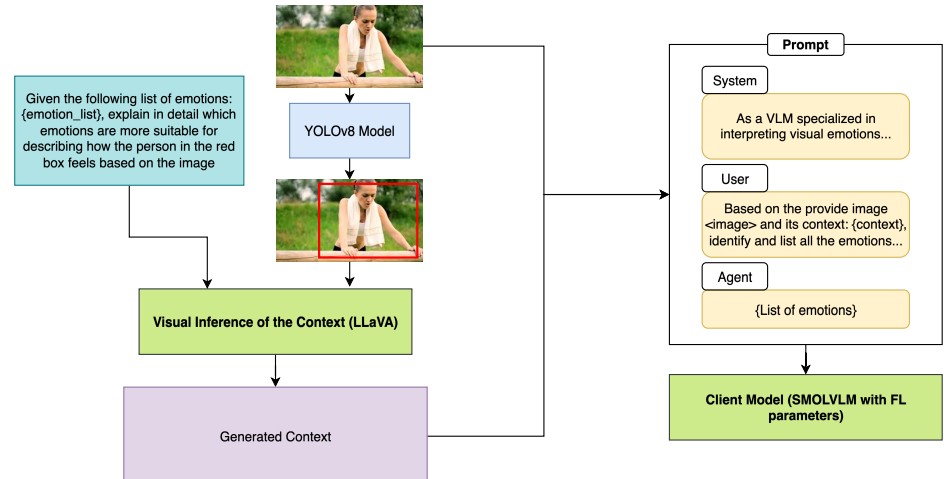

Figure 1: Phase 1: Context generation in CAREFL. YOLOv8 provides bounding boxes, which are combined with emotion prompts and passed to LLaVA to generate contextual descriptions.

## 3 CAREFL

We propose CAREFL, a framework for efficient and privacy-preserving contextual emotion recognition in images. CAREFL follows a two-phase pipeline: (i) a large vision–language model (VLM) generates rich contextual descriptions of each image, and (ii) a lightweight small VLM (SVLM) is fine-tuned with Quantized Low-Rank Adaptation (QLoRA) in a federated learning (FL) setting to perform downstream classification. The overall architecture consists of a central server and multiple clients that collaboratively optimize a global model without sharing raw data.

### 3.1 Two-Phase VLM + SVLM Pipeline

The CAREFL pipeline separates context generation from federated model training. In Phase 1, a frozen large VLM produces the descriptive context for each image. In Phase 2, this context, paired with images and serialized label sequences, are used to fine-tune an SVLM efficiently on client devices. This design transfers the cost of contextual reasoning to the large model while ensuring that client-side updates remain lightweight and privacy-preserving.

#### 3.1.1 Context generation with VLM

To enrich training data with high-level semantics, we employ LLaVA 1.5 as the context generator. Given an input image $\mathbf{X}_i^{\text{img}}$, a pre-trained YOLOv8 detector first identifies individuals via bounding boxes. These annotated regions, along with a prompt containing candidate emotions, are passed to LLaVA, which outputs a narrative-style context that describes the scene.

As illustrated in Figure 1, the generated descriptions go beyond object recognition to capture semantic and environmental cues. These textual contexts are later provided to the SVLM, enabling it to benefit from the reasoning ability of a large VLM without incurring high on-device computational costs.

#### 3.1.2 Training the SVLM for Emotion Recognition in an FL Environment

In Phase 2, the contextual descriptions are combined with the raw image and the ground-truth labels. Following the instruction fine-tuning paradigm, label sets are serialized into text sequences (`[Happy, Surprised]`), and the SVLM is trained to autoregressively generate them given the multimodal input. This naturally supports multi-label outputs while remaining compatible with the language modeling objective.

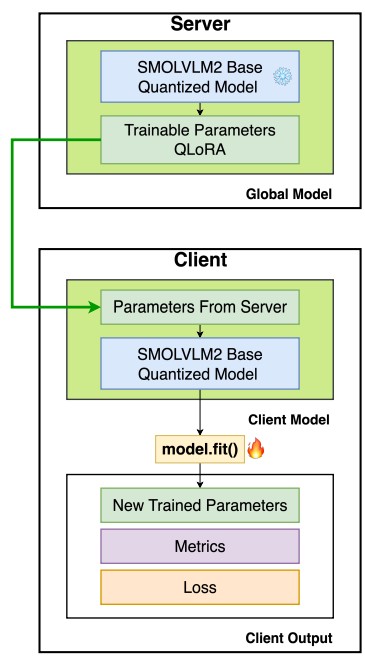 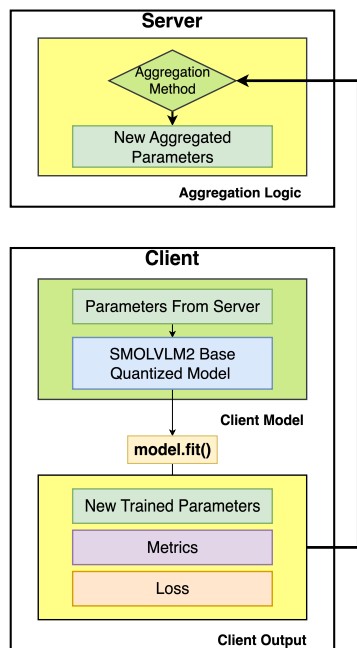

(a) Client-side fine-tuning with QLoRA.          (b) Server-side aggregation.

Figure 2: Phase 2: Federated training in CAREFL. Clients fine-tune LoRA adapters with QLoRA and send quantized updates to the server, which aggregates them into global parameters.

We adopt SMOLVLM2 as the client model due to its compact architecture and efficiency in multimodal tasks (Marafioti et al., 2025). Within CAREFL, SMOLVLM2 is fine-tuned using QLoRA: the model is quantized to 4-bit precision, base weights remain frozen, and only low-rank adapter modules are updated. This strategy minimizes memory and compute requirements, making client-side fine-tuning feasible on resource-constrained devices. Hyperparameters such as the adapter rank $r$ and scaling factor $\alpha$ are tuned to balance accuracy with efficiency.

As shown in Figure 2, the server distributes the global SVLM LoRA parameters to each client. Clients fine-tune locally on their data and return quantized adapter updates. The server aggregates these updates into new global parameters and redistributes them in the next round. By transmitting only low-rank adapter weights, CAREFL significantly reduces communication and memory overhead while maintaining accuracy, enabling scalable deployment in federated, privacy-sensitive environments.

## 4 EXPERIMENTS

All experiments were conducted using the EMOTIC dataset (Kosti et al., 2020), focused on the 26 discrete emotion labels, under the federated learning environment described in Section III. To ensure reproducibility and scalability, the Flower framework (Beutel et al., 2022) was employed as the base communication system between clients and the central server. All models were trained and evaluated on a single NVIDIA RTX 4090 GPU, with inference parameters fixed to a maximum of 256 new tokens and a repetition penalty of 1.15.

The dataset was partitioned into training (70%), validation (10%), and testing (20%), following prior works (Kosti et al., 2020; 2017). To simulate a non-IID (Non-independent and identically distributed) federated learning environment, we employed a Dirichlet partitioning strategy (Yurochkin et al., 2019) with a concentration parameter of $\alpha = 5$. This value ensures sufficient label variety across clients in the 26-class EMOTIC dataset, while lower values ($\alpha < 5$) led to unbalanced splits with empty client subsets, making training infeasible. This configuration allowed us to evaluate our

model's performance in a realistic non-IID scenario without compromising the clients' ability to learn from their local data.

Since model outputs are serialized text sequences of emotion labels, we evaluate them as multi-label text classification. Mean Average Precision (mAP), Recall, and F1-score were computed using `scikit-learn` (Pedregosa et al., 2011). These metrics capture both label coverage and correctness in the presence of multiple concurrent predictions per sample.

## 4.1 COMPARISON WITH STATE-OF-THE-ART

We benchmarked CAREFL against a range of state-of-the-art methods spanning CNN-based architectures, large vision–language models, and lightweight fine-tuning strategies. Traditional baselines include EMOTIC (Kosti et al., 2020), which fuses body and scene-context features through a two-branch CNN and the DINOv3 fine-tuned (Siméoni et al., 2025) using a structure similar to that proposed by the EMOTIC team. More recent multimodal approaches include LLaVA+CLIP (Xenos et al., 2025), which employs `clip-vit-base-patch32` with LLaVA-generated emotion-aware prompts, and NarraCap combined with GPT-4 and Mistral (Etesam et al., 2024a), where generated captions are fed to large language models without additional fine-tuning. We also evaluate GPT-4o (OpenAI et al., 2024), prompted with explicit label definitions, and LLaVA (Liu et al., 2023), tested in both zero-shot and fine-tuned settings with bounding box conditioning. Finally, we consider SMOLVLM2 (Marafioti et al., 2025), a lightweight VLM fine-tuned under the same conditions but without the contextual enrichment introduced by CAREFL. This diverse set of baselines allows us to isolate the contribution of CAREFL's two-phase design, contrasting it with both computationally heavy centralized models and lightweight federated approaches.

## 4.2 EVALUATION ON THE CAER-S DATASET

To further validate the effectiveness of CAREFL, we trained and evaluated the framework on the CAER-S dataset Lee et al. (2019), a benchmark for context-aware emotion recognition in the wild. CAER-S introduces additional challenges compared to our primary dataset, as it contains only 7 emotions, actor-based expressions, and diverse background scenes that increase intra-class variability and contextual ambiguity.

# 5 RESULTS

## 5.1 SYSTEMATIC EXPLORATION OF CAREFL

Beyond the main comparisons, we systematically evaluated the effect of key design choices on CAREFL's performance.

**Aggregation methods.** FedDyn consistently outperformed FedAvg, FedProx, and FedAdam, with best performance achieved at rank $r = 4$, $\gamma = 5.0$, and short local training using only 2 epochs. Notably, only three communication rounds were sufficient to reach peak F1-score (50.36%), indicating that CAREFL converges efficiently without requiring excessive communication.

**Client configurations.** Varying the number of clients and their participation per round revealed distinct trade-offs. With 10 clients, each holding substantial data, CAREFL achieved consistently high performance, peaking at mAP = 98.05%, Recall = 77.16%, and F1-score = 84.04% when all clients participated. In contrast, scaling to 300 clients with only 10% participation per round offered the best trade-off for large-scale FL: using just 77 images per client, CAREFL reached mAP = 96.41%, Recall = 35.74%, and F1-score = 50.14%. This result emulates realistic deployment conditions in which clients contribute small, heterogeneous datasets, capturing the practical limitations of data scarcity.

**Contextual enrichment.** Ablation studies confirmed the critical role of context: removing context reduced the F1-score to 4.74%, whereas enriching with LLaVA-generated context boosted performance to 50.36%. This highlights that contextual reasoning is the primary driver of CAREFL's improvement over lightweight federated baselines.

**Quantization.** Finally, precision experiments revealed that 4-bit QLoRA reduced memory usage substantially while maintaining competitive accuracy compared to full-precision LoRA. Centralized fine-tuning baselines also performed consistently worse than the federated CAREFL pipeline, underscoring the benefits of combining quantized adapters with federated optimization.

A more detailed discussion of hyperparameter tuning, client participation, and ablation studies is provided in Appendix A.1 and A.2.

## 5.2 PERFORMANCE ON EMOTIC

The performance of CAREFL and baseline models on the EMOTIC dataset is reported in Table 1. CAREFL consistently achieved the highest overall results, with the federated model trained using 10 clients and full participation reaching **98.05% mAP**, **77.16% Recall**, and an **F1-score of 84.04%**. This represents a substantial improvement over the fine-tuned SMOLVLM2 baseline, with a gain of +43.67 points in F1-score, underscoring the effectiveness of contextual enrichment in guiding multi-label emotion recognition. Even under the more realistic scenario of 300 clients with only 10% participation per round, CAREFL maintained strong performance with 96.49% mAP and 50.36% F1-score, outperforming larger centralized baselines such as GPT-4o, LLaVA, and EMOTIC CNN.

These results validate CAREFL's two-phase design, which offloads heavy contextual reasoning to a frozen large VLM while keeping client-side training efficient through quantized low-rank adaptation. Compared to fine-tuned LLaVA (25.03M parameters, 12 GB VRAM) and SMOLVLM2 (11.22M parameters, 10.98 GB VRAM), CAREFL reduces the number of trainable parameters to only 5.24M and memory requirements to 5.96 GB, achieving a favorable balance between efficiency and accuracy. By preserving privacy and scalability through federated training while maintaining competitive or superior accuracy, CAREFL demonstrates its suitability for deployment in real-world, resource-constrained environments.

Table 1: Performance metrics (mAP, Recall, F1-Score, and Parameters Trained) of various models compared to CAREFL. Best results in blue, and lowest in red.

| Model | mAP | Recall | F1-Score | Parameters Trained (M) | Memory (GB) |
|---|---|---|---|---|---|
| NarraCap + Mistral (Etesam et al., 2024a) | $17.61^{\pm 0.28}$ | $64.78^{\pm 0.41}$ | $23.19^{\pm 0.29}$ | N/A | N/A |
| NarraCap + GPT-4 (Etesam et al., 2024a) | $25.67^{\pm 0.32}$ | $33.16^{\pm 0.42}$ | $26.52^{\pm 0.30}$ | N/A | N/A |
| LLaVA (Zero shot) (Liu et al., 2023) | $34.21^{\pm 0.81}$ | $21.54^{\pm 0.32}$ | $22.69^{\pm 0.31}$ | N/A | 30 |
| LLaVA + CLIP (Xenos et al., 2025) | $21.96^{\pm 0.19}$ | $28.75^{\pm 0.35}$ | $17.06^{\pm 0.18}$ | 86.01 | 8 |
| GPT-4o (OpenAI et al., 2024) | $37.83^{\pm 0.87}$ | $38.53^{\pm 0.37}$ | $34.29^{\pm 0.28}$ | N/A | N/A |
| EMOTIC CNN (Kosti et al., 2020; 2017) | $25.16^{\pm 0.25}$ | $34.79^{\pm 0.42}$ | $28.67^{\pm 0.33}$ | 0.27 | 2 |
| DINOv3 FT (Siméoni et al., 2025) | $43.46^{\pm 0.19}$ | $21.73^{\pm 0.27}$ | $14.28^{\pm 0.18}$ | 20 | 12 |
| LLaVA FT + LoRA (Hu et al., 2021) | $55.69^{\pm 0.31}$ | $17.11^{\pm 0.30}$ | $23.11^{\pm 0.42}$ | 25.03 | 12 |
| SMOLVLM2 FT (Marafioti et al., 2025) | $83.78^{\pm 0.58}$ | $28.04^{\pm 0.27}$ | $40.37^{\pm 0.32}$ | 11.22 | 10.98 |
| Using 300 Clients and 10% for training | | | | | |
| CAREFL (SMOLVLM Context) | $95.17^{\pm 0.41}$ | $32.87^{\pm 0.37}$ | $46.67^{\pm 0.29}$ | 5.24 | 5.96 |
| CAREFL (LLaVA Context) | $96.49^{\pm 0.37}$ | $35.95^{\pm 0.37}$ | $50.36^{\pm 0.25}$ | 5.24 | 5.96 |
| Using 10 Clients and 100% for training | | | | | |
| **CAREFL (LLaVA Context)** | $98.05^{\pm 0.30}$ | $77.16^{\pm 0.42}$ | $84.04^{\pm 0.32}$ | 5.24 | 5.96 |

## 5.3 FEDERATED VS CENTRALIZED TRAINING

Table 2 compares centralized CAREFL against federated configurations on EMOTIC. The centralized (non-FL) setup achieved 97.01% mAP and an F1-score of 52.11. When training was distributed across 10 clients with a non-IID Dirichlet partition ($\alpha = 5$) and 100% of participation, CAREFL reached 98.05% mAP and a substantially higher F1-score of 84.04, showing that federated optimization can improve generalization by exposing models to diverse client distributions.

We selected 10 clients to approximate a moderate-scale deployment scenario, where data is distributed across multiple institutions or devices but not excessively fragmented. This number strikes a balance: too few clients would converge to centralized training with minimal heterogeneity, while too many would produce very small local datasets, leading to a more realistic scenario. Thus, 10 clients, with 100% of the participation in the training, provide a fair yet tractable benchmark for comparing federated against centralized performance. A more detailed discussion of how varying the number of clients and participation levels affects performance is provided in Appendix A.1.5.

Further, under an IID partition with 10 clients, CAREFL achieved the strongest performance overall: 98.95% mAP, 80.46 Recall, and 85.45 F1-score. The small margin between the IID and non-IID federated setups indicates that while heterogeneity introduces modest challenges, CAREFL remains robust to skewed label distributions. These results suggest that federated training not only preserves accuracy compared to centralized training but can also enhance recall and F1-score in realistic client-distributed scenarios.

Table 2: Federated vs centralized performance on EMOTIC.

| Setting | mAP | Recall | F1-score |
|---|---|---|---|
| Centralized CAREFL (IID, Non FL) | $97.01^{\pm 0.92}$ | $37.42^{\pm 0.49}$ | $52.11^{\pm 0.33}$ |
| Federated CAREFL (non-IID, $\alpha = 5$, 10 Clients 100% participation) | $98.05^{\pm 0.30}$ | $77.16^{\pm 0.42}$ | $84.04^{\pm 0.32}$ |
| Federated CAREFL (IID, 10 Clients 100% participation) | $98.95^{\pm 0.28}$ | $80.46^{\pm 0.41}$ | $85.45^{\pm 0.29}$ |

## 5.4 EVALUATION ON THE CAER-S DATASET

The performance on the CAER-S dataset is shown in Table 3. CAREFL achieves 82.33% accuracy and an F1-score of 65.49%, confirming that the framework generalizes beyond EMOTIC despite differences in label structure (single-label, seven categories). While performance is slightly lower than on EMOTIC, CAREFL remains robust, showing its adaptability to both multi-label and single-label emotion recognition tasks. This cross-dataset validation underscores the versatility of the proposed two-phase pipeline.

Table 3: Performance of CAREFL on the CAER-S and EMOTIC datasets.

| Dataset | Accuracy | mAP | Recall | F1-score |
|---|---|---|---|---|
| CAER-S | $82.33^{\pm 0.25}$ | $77.52^{\pm 0.35}$ | $64.46^{\pm 0.33}$ | $65.49^{\pm 0.30}$ |
| EMOTIC | $85.35^{\pm 0.22}$ | $96.49^{\pm 0.37}$ | $35.95^{\pm 0.37}$ | $50.36^{\pm 0.25}$ |

A comparison with existing methods on CAER-S (Table 4) further illustrates the competitiveness of CAREFL. The framework surpasses earlier context-centric approaches such as EMOTIC (74.48%), CAER-Net-S + CCIM (74.81%), and CAER-Net-S + CLEF (75.86%), and stands close to the performance of more recent specialized architectures. While state-of-the-art models like EfficientFace (85.87%) and Hybrid ConvNeXt (88.84%) achieve higher accuracy through dedicated designs tailored to single-label emotion recognition, CAREFL delivers strong results without relying on dataset-specific tuning or model redesign. This balance between generalization capability and competitive performance underscores the practical value of CAREFL as a unified, privacy-preserving emotion recognition framework that remains effective across heterogeneous datasets and task formulations.

Table 4: Comparison of CAREFL on CAER-S Dataset

| Method | Accuracy |
|---|---|
| EMOTIC (Kosti et al., 2017; 2020) | 74.48 |
| CAER-Net-S + CCIM (Yang et al., 2024a) | 74.81 |
| CAER-Net-S + CLEF (Yang et al., 2024b) | 75.86 |
| EfficientFace (Zhao et al., 2021) | 85.87 |
| Hybrid ConvNeXt (Devi et al., 2025) | **88.84** |
| **CAREFL** | 82.33 |

## 5.5 QUALITATIVE RESULTS

To illustrate CAREFL's strengths and limitations, we present three representative qualitative cases from the EMOTIC dataset. These examples highlight how contextual enrichment aids emotion recognition in diverse scenarios. The results are shown in the Figure 4.

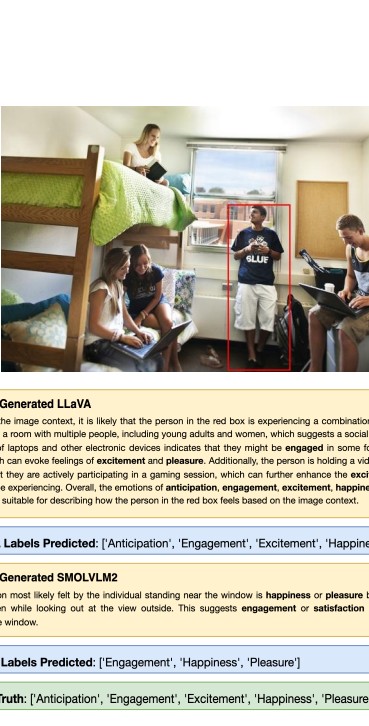

**Context Generated LLaVA**
Based on the image context, it is likely that the person in the red box is experiencing a combination of emotions. They are standing in a room with multiple people, including young adults and women, which suggests a social gathering or event. The presence of laptops and other electronic devices indicates that they might be **engaged** in some form of entertainment or work, which can evoke feelings of **excitement** and **pleasure**. Additionally, the person is holding a video game remote, which implies that they are actively participating in a gaming session, which can further enhance the **excitement** and **happiness** they may be experiencing. Overall, the emotions of **anticipation, engagement, excitement, happiness,** and **pleasure** seem to be more suitable for describing how the person in the red box feels based on the image context.

**CAREFL Labels Predicted:** ['Anticipation', 'Engagement', 'Excitement', 'Happiness', 'Pleasure']

**Context Generated SMOLVLM2**
The emotion most likely felt by the individual standing near the window is **happiness** or **pleasure** because they have their laptop open while looking out at the view outside. This suggests **engagement** or **satisfaction** with what he/she sees through the window.

**CAREFL Labels Predicted:** ['Engagement', 'Happiness', 'Pleasure']

**Ground Truth:** ['Anticipation', 'Engagement', 'Excitement', 'Happiness', 'Pleasure']

(a) Indoor scene with multiple people.

**Context Generated LLaVA**
The person in the red box appears to be experiencing a sense of **disconnection** from the rest of the room. This is evident from the fact that he is standing in front of a painting, which is a relatively solitary activity. While the presence of chairs, a dining table, and a variety of objects in the room suggests that it is a living space, the man's focus on the painting and his solitary positioning might indicate that he is not currently **enjoying** the company of others. On the other hand, the painting itself could evoke a sense of **peace** or **pleasure**, as it is an artistic expression that can provide a sense of **relaxation** or **enjoyment**. However, the emotions of **disconnection** and **peace** or **pleasure** are more suitable for describing how the person in the red box feels based on the image context."

**CAREFL Labels Predicted:** ['Disconnection', 'Peace', 'Pleasure']

**Context Generated SMOLVLM2**
The emotion "**peace**" can be inferred as he stands calmly with his eyes closed while holding an object close to him. He seems **relaxed** and at ease which suggests this feeling might have been induced by being surrounded by nature or having time alone. This could also indicate some form of meditation practice where one seeks inner peace.

**CAREFL Labels Predicted:** ['Peace']

**Ground Truth:** ['Disconnection', 'Peace', 'Pleasure']

(b) Ambiguous facial expression.

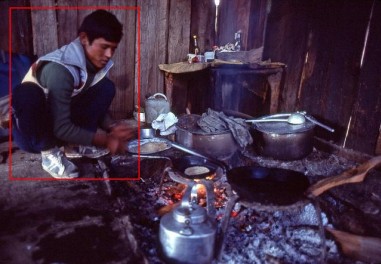

**Context Generated LLaVA**
Based on the image context, the person is **engaged** in cooking activities in a small kitchen. They are kneeling down in front of a stove, with multiple pots and pans on the counter. It appears that they are preparing a meal, possibly for a special occasion or gathering. The emotions of '**Anticipation**' and '**Confidence**' seem more suitable for describing how the person in the red box might be feeling. **Anticipation** could be due to the excitement of preparing a meal for their loved ones, while **confidence** could stem from their ability to cook and their pride in presenting their culinary skills. **Engagement** and **esteem** might not be as fitting, as the person's focus is primarily on the cooking process rather than their overall feelings about themselves or the situation.

**CAREFL Labels Predicted:** ['Anticipation', 'Confidence']

**Context Generated SMOLVLM2**
The emotion most likely felt by this individual would be **anticipation** as they seem focused intently on their task while cooking over an open fire with flames visible behind them. This suggests he has high **confidence** in themselves or others involved in the activity. They appear **engaged** due to concentration but also have **esteem** because it's clear they're taking pride in what they do.

**CAREFL Labels Predicted:** ['Anticipation', 'Confidence', 'Engagement', 'Esteem']

**Ground Truth:** ['Anticipation', 'Confidence', 'Engagement', 'Esteem']

(c) Background-driven cues.

Figure 3: Qualitative examples illustrating CAREFL's ability to leverage context for accurate emotion recognition in complex scenarios.

In the first case (indoor multi-person setting), LLaVA's detailed narrative captured activity-driven cues (gaming, social interaction), enabling CAREFL to predict the full ground truth set, while SMOLVLM context missed 'Anticipation' and 'Excitement'. This shows how a richer context enhances multi-label coverage.

In the second case (ambiguous expression), LLaVA emphasized solitude and disconnection, leading to correct predictions of 'Disconnection', 'Peace', and 'Pleasure', whereas SMOLVLM only captured 'Peace'. Here, context disambiguated emotions not visible in facial cues.

In the third case (background-driven cues), SMOLVLM's compact context captured all ground truth emotions, while LLaVA's narrower narrative led to partial predictions. This highlights the trade-off: detailed descriptions may focus too narrowly, whereas lighter summaries can generalize better.

Overall, these cases confirm that contextual enrichment enhances CAREFL's ability to infer nuanced emotions, particularly in ambiguous or low-visibility settings. At the same time, they highlight the challenge of balancing narrative detail with prediction breadth. Importantly, these qualitative findings mirror the quantitative improvements reported in Section 5.2, where context consistently boosted F1-scores (up to 50.36% and 84.05% on EMOTIC), demonstrating how context resolves ambiguity and strengthens multi-label emotion prediction.

### 5.6 LIMITATIONS

Despite its contributions, CAREFL has several limitations. First, client models were fine-tuned using QLoRA with 4-bit quantization. This configuration was selected primarily due to the limited computational resources available for this work, as it drastically reduces memory usage and communication costs. While effective for enabling federated training on resource-constrained devices, 4-bit quantization may incur a slight loss in representational capacity compared to higher-precision alternatives. Future work could explore adaptive quantization strategies or mixed-precision setups to balance efficiency with accuracy.

Second, the framework's reliance on contextual descriptions makes performance dependent on the quality of captions generated by the large VLM. Errors or biases in these descriptions may propagate to the lightweight SVLM during training. Incorporating mechanisms for context verification or noise-robust training could improve reliability.

Finally, the non-IID partitioning strategy required dataset-specific tuning. For EMOTIC, we used $\alpha = 5$ to generate realistic non-IID splits (see Section 4). For CAER-S, a smaller value of $\alpha = 0.05$ was selected due to its single-label nature, which allowed stronger heterogeneity without creating empty partitions. This difference underscores the sensitivity of Dirichlet-based partitioning to label space characteristics and highlights the need for adaptive partitioning strategies that generalize across datasets.

## 6 CONCLUSIONS

This work introduced CAREFL, a lightweight and context-aware framework for emotion recognition based on federated fine-tuning of small VLMs. By leveraging contextual descriptions from large VLMs and pairing them with emotion labels under a federated learning setup, CAREFL achieves strong performance while preserving privacy and reducing computational costs. On EMOTIC, CAREFL achieved up to 98.05% mAP and an F1-score of 84.04 under a 10-client setup with full participation, substantially outperforming baselines such as GPT-4o, LLaVA, and EMOTIC CNN. In a more realistic configuration with 300 clients and 10% participation per round, CAREFL still maintained competitive performance with 96.49% mAP and a 50.36% F1-score, showing its robustness under data fragmentation and client heterogeneity. On CAER-S, CAREFL achieved 82.33% accuracy, 77.52% mAP, 64.46% recall, and 65.49% F1-score, confirming its ability to generalize to challenging, in-the-wild conditions.

Beyond emotion recognition, CAREFL serves as a general-purpose framework: its VLM foundation makes it adaptable to other multimodal classification and description tasks where efficiency and privacy are essential. Future directions include enriching contextual reasoning with additional modalities (e.g., audio or physiological signals), mitigating data scarcity through generative augmentation, and enabling client-specific personalization in federated setups. Optimizing for heterogeneous edge devices and investigating communication-efficient aggregation strategies will further improve scalability and deployment.

In summary, CAREFL demonstrates that lightweight federated VLMs, guided by contextual enrichment and efficient fine-tuning, can serve as a practical and scalable alternative to centralized

architectures, achieving strong performance not only in controlled benchmarks but also under realistic federated scenarios with hundreds of clients and partial participation.

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

# A    Appendix

## A.1    Hyperparameter Tuning

### A.1.1    Federated Environment Optimization

CAREFL was evaluated under different client and server configurations:

- **Aggregators:** FedAvg, FedAdam, FedProx, and FedDyn with $\gamma = \{0.01, 1.0, 5.0\}$.
- **Rounds and Epochs:** Training conducted for 1, 3, 5, 7, and 10 communication rounds, with local epochs = 1, 2, 5, and 10.
- **Clients:** 10, 30, and 100 clients with non-IID partitions of EMOTIC.

The results of the hyperparameter tuning experiments are summarized in Table 5. The effects of aggregation methods, LoRA rank, number of communication rounds, and local training epochs on mAP, Recall, and F1-score are evaluated in the CAREFL pipeline. For all experiments, the best parameters are defined under a federated setting with 300 clients, where only 10% of the clients are selected per round for training. This design simulates a realistic deployment scenario in which individual clients contribute with a limited number of local examples during the training phase, reflecting practical constraints in data availability and heterogeneity. A more detailed discussion on the impact of the number of clients and examples used for training is provided in the *Effect of Number of Clients and Samples Used for Training* part.

### A.1.2    Effect of Aggregation Method and LoRA Rank

Table 5 summarizes the results of different aggregation strategies across varying LoRA ranks. At rank $r = 4$, FedDyn with $\gamma = 5.0$ achieved the best overall F1-score (49.95%) and Recall (35.89%), outperforming FedAvg and FedProx. In contrast, FedAdam achieved the lowest Recall (9.09%) and F1-score (13.78%), indicating that it is not well-suited for this task.

When increasing the rank to $r = 8$, performance remained consistent with FedDyn ($\gamma = 5.0$), which yielded the highest mAP (95.95%) and strong F1-score (49.49%). FedAvg and FedProx again showed competitive results but did not surpass FedDyn. Interestingly, FedAdam showed an improvement in Recall (20.06%) compared to $r = 4$, but its F1-score remained lower than other methods.

At $r = 16$, FedDyn with $\gamma = 5.0$ again achieved the strongest overall performance, with an F1-score of 47.10% and the highest mAP (95.75%). FedDyn with $\gamma = 1.0$ also performed competitively, achieving the highest Recall (33.21%), while FedAdam remained the weakest performer despite improving slightly in F1-score compared to $r = 4$.

These results demonstrate that the FedDyn aggregation method consistently provides the most balanced trade-off between mAP and Recall across all tested ranks, while FedAdam underperforms regardless of configuration. While a larger rank tends to improve performance in centralized setups, in our federated setting, the aggregation of a higher number of parameters actually degraded results.

Table 5: Unified comparison of results across Aggregation methods, LoRA Rank, number of rounds, and number of epochs.

| Condition | Method / Hyperparameter | mAP | Recall in percent (%) | F1-score |
|---|---|---|---|---|
| Rank r=4 | FedAdam | **90.87** | **09.09** | **13.78** |
| | FedProx | 93.23 | 27.42 | 40.02 |
| | FedAvg | 93.24 | 27.96 | 40.80 |
| | FedDyn (0.01) | 92.94 | 28.69 | 41.51 |
| | FedDyn (1.0) | 95.02 | 31.53 | 45.17 |
| | FedDyn (5.0) | 94.70 | 35.89 | 49.95 |
| Rank r=8 | FedAdam | 94.48 | 20.06 | 29.66 |
| | FedProx | 92.89 | 26.88 | 39.33 |
| | FedAvg | 92.00 | 27.67 | 40.23 |
| | FedDyn (0.01) | 92.35 | 28.07 | 40.62 |
| | FedDyn (1.0) | 91.94 | 32.26 | 45.25 |
| | FedDyn (5.0) | 95.95 | 35.19 | 49.49 |
| Rank r=16 | FedAdam | 87.30 | 12.52 | 18.73 |
| | FedProx | 92.01 | 28.40 | 41.04 |
| | FedAvg | 92.37 | 27.95 | 40.48 |
| | FedDyn (0.01) | 92.63 | 27.46 | 40.09 |
| | FedDyn (1.0) | 94.03 | 33.21 | 46.96 |
| | FedDyn (5.0) | 95.75 | 32.87 | 47.10 |
| Rounds | 1 | 86.63 | 34.75 | 47.37 |
| | 3 | 09.65 | 35.95 | 50.36 |
| | 5 | 92.74 | 33.77 | 47.60 |
| | 7 | 94.70 | 35.89 | 49.95 |
| | 10 | 94.10 | 35.15 | 49.34 |
| Epochs | 1 | 80.98 | 09.10 | 12.80 |
| | 2 | 96.49 | 35.95 | 50.36 |
| | 5 | 91.94 | 35.19 | 49.95 |
| | 10 | 92.91 | 36.06 | 50.07 |

Similarly, increasing the aggregation weight $\gamma$ for FedDyn (tested with $\gamma = 10$) did not produce improvements over the reported results.

### A.1.3 EFFECT OF COMMUNICATION ROUNDS

As observed, a single communication round produces the weakest performance (F1-score of 47.37% with an mAP of 86.63%). Increasing the number of rounds to 3 significantly improves performance, yielding the highest overall F1-score (50.36%) and Recall (35.95%).

Beyond 3 rounds, results begin to decrease. For example, with 5 rounds, the F1-score slightly drops to 47.60%, while with 7 and 10 rounds, performance remains close to the best but does not surpass it. These findings suggest that three rounds of communication are sufficient to balance efficiency and accuracy, while additional rounds provide limited benefits.

### A.1.4 EFFECT OF LOCAL TRAINING EPOCHS

With only 1 epoch, the model suffers from severe underfitting, achieving an F1-score of just 12.80% and a very low Recall of 9.10%. Increasing to 2 epochs yields the best results overall, with an mAP of 96.49% and an F1-score of 50.36%.

At 5 and 10 epochs, performance remains competitive (F1-scores around 49–50%), but does not surpass the results obtained with 2 epochs. These findings indicate that short local training (2 epochs per round) is optimal for QLoRA fine-tuning, while additional epochs provide diminishing returns and may introduce unnecessary computational cost.

### A.1.5 EFFECT OF NUMBER OF CLIENTS AND SAMPLES

The effect of varying the number of clients and their participation per communication round is summarized in Table 6. We tested three configurations: 10 clients with 2325 images each, 30 clients with 774 images each, and 300 clients with 77 images each, with participation rates of 10%, 50%, and 100%.

With 10 clients, each holding substantial data, performance remained consistently high, peaking at 98.05% mAP, 77.16% Recall, and 84.04% F1-score when all clients participated. Increasing to

30 clients reduced per-client data and overall performance, though full participation still achieved strong results (94.93% mAP, 57.28% Recall, 68.23% F1-score).

The 300-client scenario, simulating highly data-constrained devices, showed the expected drop in Recall and F1 but maintained stable mAP (96.41% at 100% participation). Interestingly, 10% participation slightly outperformed 100% in Recall (35.89% vs. 35.74%) and avoided the instability observed at 50%, where redundant or noisy updates degraded performance. This counterintuitive result highlights the sensitivity of federated setups to sampling strategies, particularly when local data availability is highly constrained.

These results illustrate the trade-off between per-client data and scalability: fewer clients yield stronger accuracy, while larger numbers reflect realistic deployment. Within CAREFL, the 300-client, 10% participation setting best balances practicality and performance, demonstrating robustness under real-world constraints. Therefore, in the CAREFL framework, the setting of 300 clients with 10% participation per round balances scalability with practical constraints and ensures the framework remains realistic for real-world applications.

Table 6: Effect of number of clients and samples used for training.

| Number of Clients | # of training images per client | % of clients used for training | mAP | Recall | F1-score |
|---|---|---|---|---|---|
| | | | in percent (%) | | |
| 10 | 2325 | 10 | 94.02 | 64.44 | 76.07 |
| | | 50 | 97.04 | 73.14 | 83.13 |
| | | 100 | 98.05 | 77.16 | 84.04 |
| 30 | 774 | 10 | 94.09 | 38.14 | 51.80 |
| | | 50 | 94.37 | 52.05 | 65.30 |
| | | 100 | 94.93 | 57.28 | 68.23 |
| 300 | 77 | 10 | 94.70 | 35.89 | 49.95 |
| | | 50 | 95.95 | 35.19 | 49.49 |
| | | 100 | 96.41 | 35.74 | 50.14 |

## A.2 ABLATION STUDIES

To evaluate the impact of different design choices in the proposed framework, CAREFL, an ablation study was conducted.

### A.2.1 IMPACT OF CONTEXT AND PROMPTING STRATEGY

The contribution of contextual enrichment to performance was measured as follows:

- **Without Context:** Image + label only.
- **Context from SMOLVLM:** Descriptions generated by the lightweight model itself.
- **Context from LLaVA:** Rich descriptions from a large VLM.
- **Prompt Variants:** Emotion categories included in (a) system prompt, (b) user prompt, or both.

The results, presented in Table 7, demonstrate that incorporating contextual information significantly improves performance. For instance, when using SMOLVLM with context and quantization, the model achieved a mAP of 95.17%, a Recall of 32.87%, and an F1-score of 46.67%. Similarly, LLaVA with context and quantization obtained the best performance across all tested configurations, reaching an mAP of 96.49%, a Recall of 35.95%, and an F1-score of 50.36%.

In contrast, removing contextual information drastically reduced the results, which dropped to only 15.01% mAP, 4.31% Recall, and 4.74% F1-score. This highlights the importance of leveraging context for emotion recognition using SVLMs.

Table 8 presents the results considering different experimental settings, including the presence of contextual information, quantization, and the different client setups.

Starting from the base SMOLVLM2 without context, quantization, or federated learning, performance is notably weak and employing an amount of 10.98 GB of VRAM. Introducing contex-

Table 7: Performance comparison by context source: mAP, Recall, and F1-Score.

| Context Source | mAP | Recall | F1-Score |
|---|---|---|---|
| None | 15.01 | 4.31 | 4.74 |
| SMOLVLM2 | 95.17 | 32.87 | 46.67 |
| LLaVA 1.5 | 96.49 | 35.95 | 50.36 |

tual enrichment yields a dramatic improvement (mAP: 77.41%, Recall: 31.06%, F1-Score: 42.54). Quantizing the model with QLoRA reduces memory usage to 5.96 GB and boosts performance further (mAP: 92.61%, Recall: 37.61%, F1-Score: 53.50).

When federated learning is enabled with 300 clients and 10% participation, the model reaches 94.70% mAP, 35.89% Recall, and 49.95% F1-score. Varying the number of clients (30 or 10, still 10% participation) sustains high mAP above 94%, while recall (38.14% and 64.44%) and F1-score (51.80 and 76.07) can increase even further depending on the client configuration and participation rate.

Overall, the table clearly demonstrates that each CAREFL component—especially context and quantization—substantially boosts recognition performance and enables scalable, efficient deployment. The federated setup, in particular, preserves high accuracy and efficiency across a range of distributed environments.

Table 8: Ablation study on the contribution of CAREFL components to performance and efficiency.

| Configuration | Context | Quantization | FL | Memory (GB) | mAP | Recall | F1-Score |
|---|---|---|---|---|---|---|---|
| | | | | | | in percent (%) | |
| Base SMOLVLM2 | ✗ | ✗ | ✗ | 10.98 | 15.01 | 4.31 | 4.74 |
| + Context | ✓ | ✗ | ✗ | 10.98 | 77.41 | 31.06 | 42.54 |
| + QLoRA | ✓ | ✓ | ✗ | 5.96 | 92.61 | 37.61 | 53.50 |
| + Federated Environment (300 Clients, 10% participation) | ✓ | ✓ | ✓ | 5.96 | **94.70** | 35.89 | 49.95 |
| + Federated Environment (30 Clients, 10% participation) | ✓ | ✓ | ✓ | 5.96 | 94.04 | 38.14 | 51.80 |
| + Federated Environment (10 Clients, 10% participation) | ✓ | ✓ | ✓ | 5.96 | 94.02 | **64.44** | **76.07** |

Table 9: Evaluation of QLoRA quantization and centralized fine-tuning benchmarks.

| Setting | mAP | Recall | F1-score |
|---|---|---|---|
| | | in percent (%) | |
| **QLoRA Quantization Evaluation** | | | |
| Full Precision Fine-tuning (LoRA) | 92.39 | 42.41 | 54.71 |
| 4-bit QLoRA Fine-tuning | 96.49 | 35.95 | 50.36 |
| | | | |
| **Centralized Fine-tuning Benchmarks** | | | |
| Full Fine-tuning (non-FL) | 77.41 | 31.06 | 42.54 |
| Centralized QLoRA Fine-tuning | 92.61 | 37.61 | 53.50 |

### A.2.2 QLoRA Quantization Evaluation

The effect of quantization on SMOLVLM was assessed:

- **Full Precision Fine-tuning**: LoRA adapters without quantization.

- **4-bit QLoRA Fine-tuning**: LoRA adapters with quantization.

Table 9 presents the results of evaluating the effect of quantization on the model SMOLVLM2. The full precision LoRA fine-tuning achieved an mAP of 92.39%, Recall of 42.41%, and F1-score of 54.71%. When applying 4-bit QLoRA, the mAP increased to 96.49%, showing that quantization not only reduces the memory footprint but also improves performance in terms of retrieval precision. However, Recall dropped to 35.95%, and F1-score slightly decreased to 50.36%, indicating that quantization introduces a trade-off: while overall accuracy is improved, sensitivity to correctly identifying all relevant instances is reduced.

### A.2.3 Centralized Fine-tuning Benchmarks

Two centralized settings were tested to contextualize the federated results:

- **Full Fine-tuning (non-FL)**: Complete model training without QLoRA.
- **Centralized QLoRA Fine-tuning**: Matching federated setup but with centralized dataset.

The centralized fine-tuning benchmarks are also reported in Table 9. Full fine-tuning (non-FL) reached 77.41% mAP, 31.06% Recall, and 42.54% F1-score, which is notably lower than both federated and quantized settings. Centralized QLoRA fine-tuning improved these results, achieving 92.61% mAP, 37.61% Recall, and 53.50% F1-score. This confirms the benefits of QLoRA even outside the federated context, outperforming traditional centralized training. Compared to the federated setup, centralized QLoRA attains competitive results, but the federated CAREFL pipeline provides superior scalability and robustness against data heterogeneity while maintaining higher overall precision.

### A.3 Training Efficiency and Resource Analysis

To complement the performance comparison in Section IV, Table 10 reports detailed training efficiency metrics for all baseline models and the proposed CAREFL framework. The table includes memory requirements, dataset partitioning, training time per client, total wall-clock time, and normalized training time per sample.

Table 10: Training efficiency and resource analysis

| Model | Memory MB | Memory GB | Samples | Samples by Client | Batch Size | Local Epocs | Training Time Per Client | Total Clients | Sampled Clients | FL Rounds | Total Training Time (s) | Total Training Time (hh:mm:ss) | Training Time Per Sample (s) |
|---|---|---|---|---|---|---|---|---|---|---|---|---|---|
| LLaVA + CLIP | 8192 | 8.00 | 23265 | - | 64 | 50 | 21744 | - | - | - | 21744 | 06:02:24 | 0.019 |
| EMOTIC | 2047 | 2.00 | 23265 | - | 52 | 15 | 15053 | - | - | - | 15053 | 04:10:53 | 0.043 |
| LLaVA FT + LoRA | 12288 | 12.00 | 23265 | - | 16 | 4 | 81252 | - | - | - | 81252 | 22:34:12 | 0.873 |
| SMOLVLM FT | 11240 | 10.98 | 23265 | - | 16 | 4 | 71894 | - | - | - | 71894 | 19:58:14 | 0.773 |
| CAREFL (SMOLVLM Context) | 6098 | 5.96 | 23265 | 77 | 4 | 3 | 542 | 300 | 30 | 3 | 48780 | 13:33:00 | 0.699 |
| CAREFL (LLaVA Context) | 6098 | 5.96 | 23265 | 77 | 4 | 3 | 558 | 300 | 30 | 3 | 50220 | 13:57:00 | 0.720 |

The training efficiency analysis highlights substantial differences in resource requirements and computational costs across baseline models and the proposed CAREFL framework. Among centralized baselines, EMOTIC is the most lightweight, requiring only 2 GB of memory and achieving the lowest training time per sample (0.043 s), though at the expense of lower overall performance. In contrast, LLaVA FT and SMOLVLM FT exhibit significantly higher computational demands, consuming 11–12 GB of memory and exceeding 70,000 seconds of total training time, with per-sample training times of 0.773–0.873 s, underscoring their limited practicality in resource-constrained environments. CAREFL, in contrast, achieves a favorable compromise by distributing training across 300 clients with different data partitions. Each client requires only 6 GB of memory and achieves per-sample training times of 0.699–0.720 seconds, resulting in overall training durations of approximately 13–14 hours. This decentralized efficiency highlights CAREFL's scalability, as the framework enables resource-constrained clients to participate in training without sacrificing performance, while mitigating the prohibitive costs of centralized fine-tuning.

## B Qualitative Error Analysis

This section analyzes representative failure cases from the model's predictions, where the CAREFL framework incorrectly assigned emotion labels. Each figure pairs the automatically generated contextual description (using LLaVA), predicted emotion labels, and ground-truth annotations.

In Figure 4a, the model predicted 'Sorrow' and 'Grief' based on contextual cues of mourning and collective solemnity. The ground truth, however, was 'Anticipation', 'Confidence', and 'Engagement', reflecting the individual's active role or focus rather than overt emotional distress. The overreliance on scene-level cues for sorrow led to a mismatch, in this case, the LLaVA-generated context, which automatically annotated the real emotions as "*not suitable to describe the man's emotions*", illustrating the challenges of context-aware emotion recognition, where the emotional state may diverge from the general atmosphere.

Figure 4b shows the case where the predicted labels were 'Excitement' and 'Joy', influenced by cues of group engagement and the apparent enjoyment of play. However, the annotated ground

truth was 'Disquietment', 'Engagement', and 'Fatigue'. The model missed cues of social fatigue and discomfort, perhaps because these emotions are less visually salient or contextually masked by overall engagement, underscoring the challenge of distinguishing between diverse social affects.

Finally, in Figure 4c, the model predicted 'Emotion', 'Exhilaration', 'Enthusiasm', and 'Eagerness' as the most probable emotions for the highlighted participant. While the LLaVA-generated context effectively describes traits like engagement and excitement, the ground truth includes 'Aversion', 'Confidence', 'Disquietment', 'Engagement', and 'Excitement'. The model failed to capture the emotional mix of aversion and disquietment, likely due to ambiguous body language and group dynamics, highlighting the difficulty of modeling subtle interpersonal signals and negative affect in active scenes.

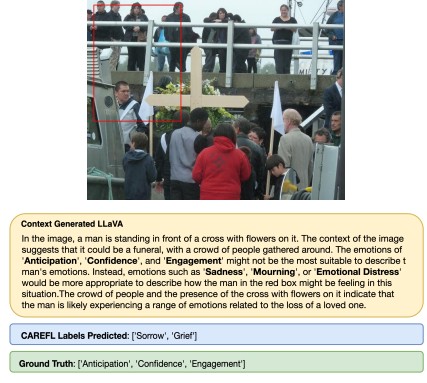

(a) Emotional state diverges from the general atmosphere

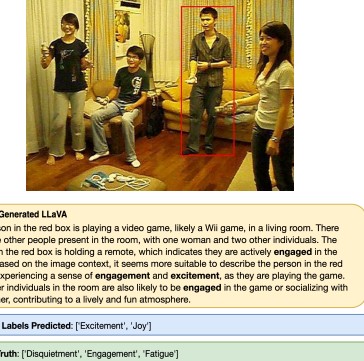

(b) Challenges in diverse social affects

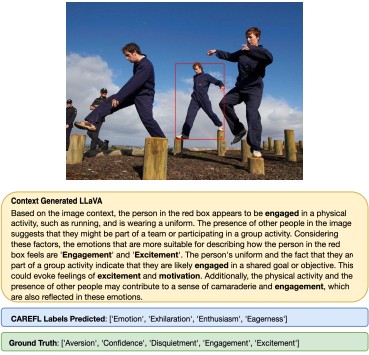

(c) Ambiguous body language and group dynamics

Figure 4: Qualitative examples illustrating CAREFL's inference errors.

## C HYPOTHESIS TESTS

To rigorously assess the differences among the evaluated models, we conducted hypothesis tests across the three performance metrics: mean Average Precision (mAP), Recall, and F1-Score. The Friedman test results for all three metrics yielded large test statistics accompanied by highly significant p-values ($\approx 10^{-7}$ to $10^{-8}$), indicating rejection of the null hypothesis of equivalent model performance. Following this, post-hoc pairwise comparisons with the Nemenyi test revealed that the CAREFL variants consistently form a statistically superior cluster relative to the NarraCap-based baselines, LLaVA variants, and conventional CNN or DINOv3 models. This is clearly evidenced by the Nemenyi heatmaps and critical difference diagrams (Figures 6 and 5), where CAREFL models rank significantly higher and demonstrate consistent, statistically robust improvements across all metrics. Conversely, other baselines such as NarraCap + Mistral, NarraCap + GPT-4, and certain LLaVA configurations exhibit no statistically significant differences from each other, as reflected by overlapping rank intervals and non-significant heatmap regions. Collectively, these results provide support that context-aware federated learning, as instantiated by CAREFL, offers reliable and generalizable performance gains in complex, reliability-critical emotion recognition tasks, substantiating the efficacy of our proposed architectural design.

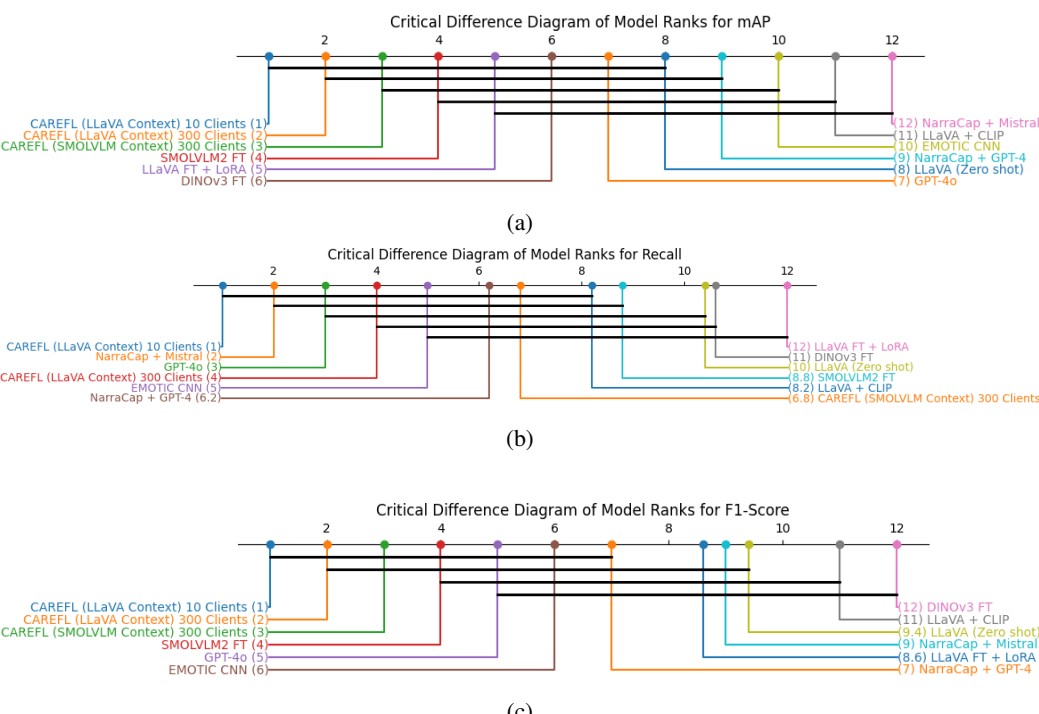

Figure 5: Critical Difference Diagram for each metric

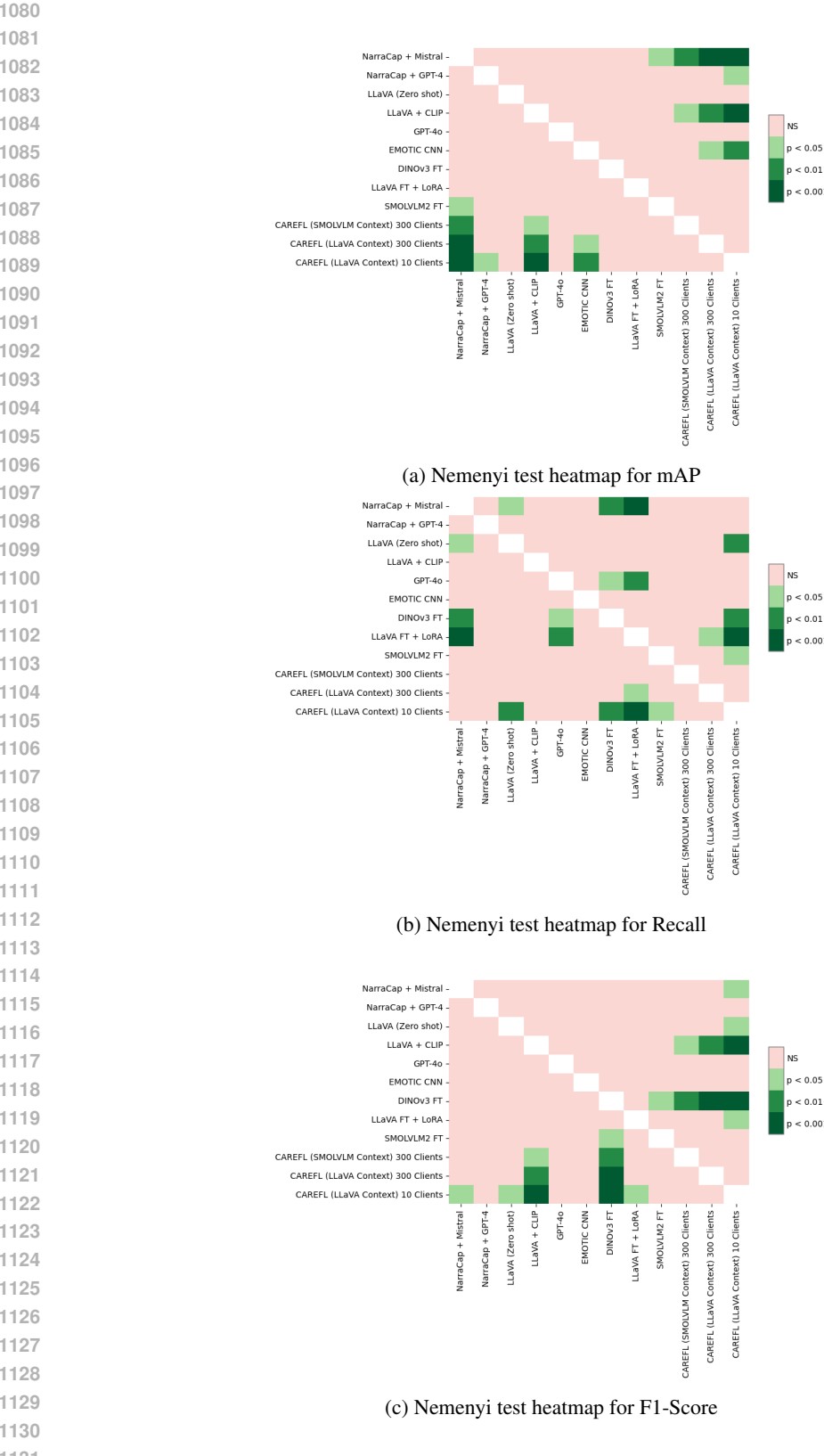

(a) Nemenyi test heatmap for mAP

(b) Nemenyi test heatmap for Recall

(c) Nemenyi test heatmap for F1-Score

Figure 6: Nemenyi test heatmaps.

