# OpenReview forum: "CAREFL: Context-Aware Recognition of Emotions with Federated Learning"
_ICLR.cc/2026/Conference — ICLR 2026 Conference Withdrawn Submission_

### Official Review · Reviewer_DFpT · 2025-10-25

**Soundness:** 2
**Presentation:** 1
**Contribution:** 1
**Rating:** 2
**Confidence:** 4

**Summary:**

The paper proposes a federated learning framework for emotion recognition from images, designed to balance contextual reasoning, privacy, and computational efficiency. The system operates in two stages: (1) a large vision–language model (LLaVA 1.5) generates contextual captions for each image, and (2) a lightweight vision–language model (SMOLVLM2) is fine-tuned with Quantized Low-Rank Adaptation (QLoRA) in a federated setting. This design enables decentralized training without sharing raw data while leveraging semantic context from the larger model. Experiments on EMOTIC and CAER-S datasets show that CAREFL achieves higher mean average precision and F1-scores compared to larger centralized models such as GPT-4o and LLaVA, while reducing memory usage and model size.

**Strengths:**

The paper’s contributions include: (1) proposing a novel two-phase federated framework combining large-model context generation with small-model adaptation, (2) introducing an efficient QLoRA-based fine-tuning scheme for lightweight federated training, and (3) comparative and ablation studies across datasets, client numbers, aggregation methods, and quantization settings.

**Weaknesses:**

Despite its technical framing, the paper appears conceptually weak and executionally shallow:

(1) The link between “context awareness” and federated learning is not clearly articulated. Context generation is performed offline using an existing large model, not integrated dynamically into the FL process. This makes the “context-aware” claim superficial.

(2) Illustrations and explanation lack clarity. Figures 1 and 2 are schematic and omit crucial architectural or algorithmic details; the paper mostly reuses known components (YOLO, LLaVA, QLoRA) with limited methodological innovation.

(3) Evaluations rely on narrow datasets (EMOTIC, CAER-S) without broader benchmarking or significant statistical analysis; performance comparisons against massive centralized models seem to be not fair and lack deeply analyzed.

(4) Many claims (e.g., “context improves emotion recognition”) are intuitive but not theoretically supported or quantitatively dissected.

Overall, presentation feels more like a system demonstration than a rigorous ICLR-level contribution; key insights or innovations are missing.

**Questions:**

1. How exactly does “context awareness” influence the federated learning process? Does context affect model aggregation or only data preprocessing? Why was context generation performed offline instead of integrated dynamically during training?

2. How does the framework generalize to other tasks beyond emotion recognition?

3. How are biases or errors from LLaVA-generated captions mitigated during federated fine-tuning?

---

> ### Author Response · Authors · 2025-11-20
> **Official Comment by Authors**
>
> Thank you for the review. We respond to your comments on weaknesses and questions below.
>
> Weakness 1
> > The link between “context awareness” and federated learning is not clearly articulated. Context generation is performed offline using an existing large model, not integrated dynamically into the FL process. This makes the “context-aware” claim superficial.
>
> As only inference of the VLM (LLaVa) is required, less computational resources will be needed on the client’s side for training. We show in Table 8 the amount of memory that is needed for fine-tuning, which is 5,96 GB VRAM for CAREFL and possible with consumer GPUs. The inference with the large VLM is only required once during the training phase, as the generated context remains constant.
>
> Weakness 2
> > Illustrations and explanation lack clarity. Figures 1 and 2 are schematic and omit crucial architectural or algorithmic details; the paper mostly reuses known components (YOLO, LLaVA, QLoRA) with limited methodological innovation.
>
> We integrate components in a novel way to achieve superior performance in a challenging domain and benchmark. We understand that integration might be regarded as a lack of methodological innovation, but we think that this line is not so clear. Similar claims could be made about generative adversarial networks, an idea that only integrates two competing networks to train them with a novel loss.
> We think that it is important to show that we can effectively combine fine-tuning, federated learning, and inference of larger, pre-trained models to effectively solve a challenging vision problem in resource-constrained environments.
>
> Weakness 3
> > Evaluations rely on narrow datasets (EMOTIC, CAER-S) without broader benchmarking or significant statistical analysis; performance comparisons against massive centralized models seem to be not fair and lack deeply analyzed.
>
> (1) We chose two complementary datasets (EMOTIC, context-rich vs. CAER-S, facial-only) to validate the model’s adaptability. As stated in the future work section, we plan to expand the modalities and domains by incorporating speech and video.
>
> (2) Given the computational requirements and our resources, we cannot guarantee statistical significance at this stage as this would require many repetitions of the experiments. However, we updated the results in Tables 1-3 to include 1-sigma (standard deviation) intervals over model evaluations and have added a hypothesis test in Appendix C.
>
> (3) In our opinion, a massive centralized model should give the best possible results for any dataset at the cost of compute and a lack of privacy. Hence, we compared our approach to large pretrained models (LLaVa 1.5, GPT4o), approaches that use prompt engineering to help those massive models (Narracap with Mistral and GPT4), and custom state-of-the-art models trained (EMOTIC) or fine-tuned (DINOv3) for this task.
>
> Weakness 4
> > Many claims (e.g., “context improves emotion recognition”) are intuitive but not theoretically supported or quantitatively dissected.
>
> We did ablation studies to confirm that rich context helps. The results are reported in Table 8 (Appendix A.2). The mAP increases from 15.01% to 77.41% just with the context and up to 94.70% in the federated environment.
> We updated the manuscript replacing Table 6, and breaking the results in 2 new tables, Table 7 just talking about the context, and Table 8 talking about the ablation of the components.
>
>
>
> Question 1
> > How exactly does “context awareness” influence the federated learning process? Does context affect model aggregation or only data preprocessing? Why was context generation performed offline instead of integrated dynamically during training?
>
> Context generation was performed offline to ensure efficiency, reproducibility, and privacy preservation. Dynamically invoking large vision-language models (like LLaVA) during training would substantially increase computational overhead.
>
> Question 2
> > How does the framework generalize to other tasks beyond emotion recognition?
>
> Since CAREFL design is modular, replacing the downstream head and adjusting task-specific prompts enables straightforward adaptation without altering the core training protocol.
>
> Question 3
> > How are biases or errors from LLaVA-generated captions mitigated during federated fine-tuning?
>
> We acknowledge that the current work does not deeply address potential biases or errors introduced by LLaVA-generated captions. While basic filtering and preprocessing were applied to remove obviously inconsistent or irrelevant captions, a comprehensive analysis of bias propagation was beyond the scope of this study. We agree that exploring bias mitigation strategies during federated fine-tuning, such as adaptive weighting of contextual inputs, bias-aware aggregation, or self-corrective context refinement, represents an important avenue for future work.

---

### Official Review · Reviewer_7R4H · 2025-10-31

**Soundness:** 2
**Presentation:** 3
**Contribution:** 2
**Rating:** 4
**Confidence:** 3

**Summary:**

The paper presents CAREFL, a framework designed for efficient and privacy-preserving emotion recognition. First, a large vision-language model (LLaVA 1.5) generates contextual descriptions of images to enrich semantic information. Second, a lightweight model (SMOLVLM2) is fine-tuned using QLoRA within a federated learning setup. This method allows distributed training without sharing raw data. Experiments on the EMOTIC and CAER-S datasets show that CAREFL achieves high accuracy and F1-scores while significantly reducing computational and memory requirements.

**Strengths:**

1. The two-phase design cleverly combines large VLMs for context generation with lightweight models for federated learning is reasonable.

2. Experiments show strong performance, surpassing larger centralized models like GPT-4o and LLaVA.

3. The paper is well-written and easy to read.

**Weaknesses:**

1. The proposed two-phase design relies on rich contextual descriptions generated offline using LLaVA 1.5. However, in real-world or real-time emotion recognition scenarios, such offline pre-generation is impractical due to latency, computational overhead, and privacy constraints. This is inconsistent with the author's claim.

2. The experimental setup overlooks realistic aspects of federated learning, such as heterogeneous client data distributions, communication latency, and device variability.

3. Could you show examples of successful and failed predictions for a discussion?

**Questions:**

Please see Weaknesses.

---

> ### Author Response · Authors · 2025-11-20
> **Official Comment by Authors**
>
> Thank you for the review. We respond to your comments on weaknesses and questions below.
>
> Weakness 1
> > The proposed two-phase design relies on rich contextual descriptions generated offline using LLaVA 1.5. However, in real-world or real-time emotion recognition scenarios, such offline pre-generation is impractical due to latency, computational overhead, and privacy constraints. This is inconsistent with the author's claim.
>
> The strength of the two-phase design is mainly that it makes *training* on clients with limited resources (e.g., VRAM) possible. We get a rich context from a large pretrained model, and we can finetune a smaller model to use the context information to make more accurate predictions for our classification task.
>
> Weakness 2
> > The experimental setup overlooks realistic aspects of federated learning, such as heterogeneous client data distributions, communication latency, and device variability.
>
> Regarding the realism of the federated setup, we acknowledge that real-world systems present additional challenges such as device variability and communication delays. However, our experimental design intentionally incorporates the key source of complexity in practical federated deployments: heterogeneous and imbalanced client data distributions (Table 2).
>
> To model this, we partition each dataset using a Dirichlet distribution, where the \alpha parameter controls the degree of non-IID behavior among clients. For the EMOTIC dataset, we used $\alpha = 5$, which produces moderately heterogeneous but still diverse label distributions across clients — reflecting environments where users share overlapping emotion categories but still exhibit individual variation.
>
> For CAER-S, we used $\alpha = 0.05$, a much smaller value that creates highly skewed, disjoint, and strongly non-IID partitions. This setting intentionally stresses the model by assigning clients very narrow subsets of labels, simulating challenging real-world cases where users or devices capture only specific types of emotions or scenarios.
>
> We evaluated CAREFL with 10, 30, and 300 clients, including 10%, 50% and 100% participation per round. With these different participation rates, we create natural variability in label coverage and update frequency. This partial participation mimics asynchronous device availability in mobile or edge deployments. Our experiments showed that CAREFL remains stable even under these severe non-IID conditions (Appendix A.1.5).
>
> While we acknowledge that additional factors, such as network latency and hardware variability, can further affect real-world systems, the combination of Dirichlet-based partitions, a large client population, and partial participation provides a strong and realistic approximation of federated learning behavior in practice.
>
> Weakness 3
> > Could you show examples of successful and failed predictions for a discussion?
>
> We added Appendix B with 3 additional examples of failed predictions.

---

> > ### Comment · Reviewer_7R4H · 2025-11-22
> >
> > Thank you for the detailed response. I still think this paper to be incremental. Therefore, I maintain my view that the paper is borderline and would not mind if paper is accepted. I will keep the score.

---

### Official Review · Reviewer_Z2WH · 2025-10-31

**Soundness:** 3
**Presentation:** 3
**Contribution:** 3
**Rating:** 6
**Confidence:** 3

**Summary:**

The paper proposes CAREFL, a two-phase framework for multimodal emotion recognition that (1) uses a large frozen VLM (LLaVA-1.5) offline to generate rich scene/subject contextual descriptions, and (2) federatedly fine-tunes a small, efficient SVLM (SMOLVLM2) on client devices using quantized low-rank adapters (QLoRA). Experiments on EMOTIC (multi-label) and CAER-S (7 classes) show large gains in mAP and varying gains in F1/Recall.

**Strengths:**

1. This paper proposed a light-weight training approach, which is shown to be effective at achieving promising model performance.
2. This paper conducted comprehensive evaluation which covers different aggregation algorithms (FedDyn, FedAvg, FedProx, FedAdam), LoRA ranks, quantization settings (4-bit QLoRA vs full LoRA)
3. Large performance improve on EMOTIC benchmark

**Weaknesses:**

1. Claims of outperforming huge baselines need more careful parity. The paper states CAREFL outperforms GPT-4o, LLaVA and other heavy models — but many of these baselines are used in zero-shot or prompting setups while CAREFL is fine-tuned (and in federated settings).
2. Lack of evaluation benchmarks. The proposed models and baselines are mostly evaluated on EMOTIC. The results of the proposed model on CAER-S are not compared with any baselines.

**Questions:**

1. For results on EMOTIC, why mAP is so high while the recall and F1 are modest?

---

> ### Author Response · Authors · 2025-11-20
> **Official Comment by Authors**
>
> Thank you for the review. We respond to your comments on weaknesses and questions below.
>
> Weakness 1
> > Claims of outperforming huge baselines need more careful parity. The paper states CAREFL outperforms GPT-4o, LLaVA and other heavy models — but many of these baselines are used in zero-shot or prompting setups while CAREFL is fine-tuned (and in federated settings).
>
> Indeed, large multimodal models such as GPT-4o and LLaVA were evaluated in zero-shot or prompt-based configurations, whereas CAREFL underwent fine-tuning within a federated learning setup. This difference aligns with our core motivation: to demonstrate that task-specialized fine-tuning of lightweight VLMs can reach or even surpass the performance of large-scale, general-purpose models while maintaining privacy and efficiency. Our intent is not to claim universal superiority, but to highlight that when fine-tuned in realistic, distributed conditions, small models can achieve competitive performance at a fraction of the computational and data privacy cost. This is particularly relevant for emotionally sensitive applications such as education or therapy, where local fine-tuning is both feasible and necessary.
>
> Weakness 2
> > Lack of evaluation benchmarks. The proposed models and baselines are mostly evaluated on EMOTIC. The results of the proposed model on CAER-S are not compared with any baselines.
>
> EMOTIC is a very challenging dataset because it asks models to recognize emotions in real-life images that contain a lot of variety and complexity. The dataset requires understanding not just faces, but also body language and the surrounding environment. Often, parts of the body or face are hidden or turned away, making it hard for algorithms to find emotional clues. In addition, there are 26 different emotion labels, and each case may show more than one emotion at a time. All these factors mean models must learn to use small hints from body pose, scene context, and subtle differences between emotions, like the one that we are proposing. This makes EMOTIC much harder than simpler datasets that just use clear facial expressions or only a few emotions, like the FER, FER+, and CAERS. We added 3 new examples to the appendix to show how challenging the dataset is, and we also included Table 4 showing the comparison with other baselines using the CAER-S dataset.
>
> Question 1
> > For results on EMOTIC, why mAP is so high while the recall and F1 are modest?
>
> The observed difference between mAP and Recall/F1-Score in multi-label emotion datasets like EMOTIC is a direct consequence of class imbalance, the data distribution of the EMOTIC dataset is as follows: Engagement constitutes 50% of the samples, whereas emotions such as Anticipation (27%), Happiness (26%), Excitement (26%), and Confidence (23%) are also well represented. In contrast, minority emotions such as Aversion (2%), Anger (1%), Fear (1%), Pain (1%), and Embarrassment (1%) are significantly less frequent. Metrics like mAP are less affected by this imbalance, leading to higher scores, while Recall and F1-Score drop due to underrepresented classes. The reported results were obtained under a simulated real-world federated environment involving 300 clients, each trained with limited local data, and with only 10% of clients sampled per communication round. This setup inherently increases data sparsity and variability across rounds, leading to lower recall and F1. When evaluated under an idealized setting with 10 clients and full (100%) participation, CAREFL consistently achieved higher scores across all metrics, confirming that the model’s capacity is not the limiting factor but rather the realistic federated training dynamics.

---

### Author Response · Authors · 2025-11-21
**Global Response**

Thank you to all reviewers for their constructive feedback.
The revised manuscript includes the following **new sections, updated tables, and added analyses**:


### **1. New Baseline Comparison on CAER-S** *(Z2WH)*

* Added **Table 4**, providing a direct comparison of CAREFL against CAER-S baselines.
* Addresses the request for broader benchmarking beyond EMOTIC.

### **2. New Appendix B: Failed Prediction Examples** *(Z2WH, 7R4H)*

* Added **three additional failed prediction examples** in **Appendix B**, with detailed explanations.
* Improves qualitative analysis and failure-case interpretation.

### **3. Updated Tables 1–3 With Statistical Measures** *(DFpT)*

Tables **1**, **2**, and **3** now include:

* **1-sigma standard deviation intervals** across multiple runs
* A **new hypothesis test** added in **Appendix C** to assess statistical significance

These changes directly address the request for deeper statistical analysis.

### **4. Added DINOv3 Fine-Tuning Results** *(Z2WH, 7R4H, DFpT)*

* **Table 1** now includes **DINOv3 fine-tuning results**, expanding the comparison with backbone architectures.

### **5. Reorganized Context and Ablation Sections** *(Z2WH, 7R4H, DFpT)*

* The original **Table 6** was replaced and divided into two clearer tables:

  * **Table 7:** Results focused exclusively on the effect of *context*.
  * **Table 8:** Ablation study analyzing the contribution of each CAREFL component.
* This improves clarity and directly addresses concerns about interpretability and component attribution.

We hope these revisions address the concerns raised by the reviewers and clarify the strengths, generality, and practical relevance of CAREFL. We appreciate the constructive feedback and are happy to revise further if needed.

---

### Note · Authors · 2026-01-26

I have read and agree with the venue's withdrawal policy on behalf of myself and my co-authors.

---

### Meta-Review · Area_Chair_s9SB · 2025-12-30

**Summary:**

This paper was reviewed by three experts and received 6, 4, 2 as the initial ratings. The reviewers agreed that the paper proposes a lightweight training scheme combining large-model context generation with small-model adaptation, the experimental results are encouraging, and that the paper is well-written and easy to read.

Reviewer DFpT has mentioned that the proposed method mostly re-uses existing techniques with little methodological innovation. In their response, the authors have mentioned that they have effectively combined fine-tuning, federated learning, and inference of large, pre-trained models to solve a vision problem, which was shown to achieve superior performance on a challenging benchmark. While this is useful and has merit, the AC feels that this cannot be considered as a substantially novel contribution, given the standards of ICLR.

Reviewers Z2WH and DFpT raised concerns about the comparison baselines and mentioned that comparing against massive foundation models in the zero-shot setup, without fine-tuning may not be fair. In their response, the authors have clarified their motivation, and have included additional comparison baselines for the CAER-S dataset (in Table 4). The results demonstrate that state-of-the-art models like EfficientFace and Hybrid ConvNeXt outperform the proposed method. The authors have explained this by saying that these models achieve higher accuracy through dedicated designs tailored to single-label emotion recognition, while CAREFL delivers strong results without relying on dataset-specific tuning or model redesign. However, this is not a very convincing argument and the large difference in accuracy (greater than 6%) between the proposed CAREFL and Hybrid ConvNeXt remains a concern.

Reviewers Z2WH and DFpT also raised concerns about narrow evaluation benchmarks, without broader benchmarking. While the authors have explained that they have used two complementary datasets (EMOTIC, context-rich vs. CAER-S, facial-only), the AC feels that a more thorough validation on other popular emotion recognition benchmarks can further validate the usefulness of the proposed method.

During the post-rebuttal discussion period, Reviewer 7R4H mentioned that the reviewer thinks that the contributions of the paper are incremental, and has maintained the rating of 4.

We appreciate the authors' efforts in meticulously responding to each reviewer’s comments, and conducting the additional experiments to answer some of the reviewers' questions (such as including the additional comparison baselines for the CAER-S dataset, including the DINOv3 fine-tuning results in Table 1, and including the standard deviation values in the tables). However, in light of the above discussions, we conclude that the paper may not be ready for an ICLR publication in its current form. While the paper clearly has merit, the decision is not to recommend acceptance. The authors are encouraged to consider the reviewers' comments when revising the paper for submission elsewhere.

**Reviewer Concerns:**

Please see my comments above.

**Reviewer Scores:**

Reviewer Z2WH -> 6

Reviewer 7R4H -> 4

Reviewer DFpT -> 3

---

### Decision · Program_Chairs · 2026-01-26

Reject